# Sensory trial of camel milk powder among pastoralist communities of the Somali Region, Ethiopia

Ahmed Mohammed Ibrahim[1,2,3]*, Seid Mohammed Ali[1,2,3], Yahya Maidane Osman[3], Fathiya Budul Ismail[3], Mohamed Omar Osman[3,4], Pauline Rouchon[5], Mukhtar Harir Hussein[1,2], Abdifatah Muktar Muhummed[3,6], Ramadan Budul Yusuf[3], Raymond Place[5], Jan Hattendorf[1,2], Rea Tschopp[1,2,7], Pascal Mäser[1,2], Jakob Zinsstag[1,2]

**1** Swiss Tropical and Public Health Institute, Kreuzstrasse 2, Allschwil, Switzerland, **2** University of Basel, Petersplatz 1, Basel, Switzerland, **3** Institute of health sciences, Jigjiga University, Jigjiga Ethiopia, **4** Department of Public Health, College of Medicine and Health Sciences, Hawassa University, Hawassa, Ethiopia, **5** School of Agricultural, Forest and Food Sciences, Bern University of Applied Sciences (BFH-HAFL), Zollikofen, Switzerland, **6** Armauer hansen research institute, Addis Ababa, Ethiopia, **7** Faculty of Medicine and Biology, University of Lausanne, Vaud, Switzerland

* ahmey114baba@gmail.com, ahmedmohammed@jju.edu.et, ahmed.ibrahim@swisstph.ch, ahmed.ibrahim@unibas.ch

## Abstract

### Background

Camel milk is a vital source of nutrition for people living in many arid and semi-arid regions. Camel milk powder offers a valuable opportunity for the dairy industry to develop and launch innovative products in the milk and dairy market.

### Objective

To evaluate the sensory preference of camel milk powder compared to fresh camel milk among pastoralist communities in the Somali Region of Ethiopia.

### Methods

A single-blinded sensory crossover study was conducted among randomly selected 248 pastoralist communities in the Somali Region. Data were collected using structured questionnaires. Participants did not know whether they tasted fresh milk or camel milk powder. The sequence was randomized, assigning either fresh/camel milk powder or powdered/fresh camel milk to each participant. Data was summarized using mean, median, frequency, and percentage and was presented using charts and tables. Bi-variable and multivariable binary logistic regression were used to identify variables associated with the outcome. Statistical associations were assessed using odds ratios and 95% confidence intervals, with significance declared at a P-value < 0.05. A t-test was used to compare milk spending and milk liking levels between powdered and fresh camel milk.

**Data availability statement:** All relevant data are within the paper and its Supporting Information files.

**Funding:** The author(s) received no specific funding for this work.

**Competing interests:** The authors have declared that no competing interests exist.

## Results

In this study, 31% (95% CI: 25.5%–37.1%) preferred powdered milk among pastoralists in the Somali Region. Factors significantly associated with preference of camel milk powder included being an urban pastoralist resident, AOR = 2.02 (95% CI: 1.30, 3.16), and being female, AOR = 1.94 (95% CI: 1.25, 3.01). There is no statistically significant mean difference between fresh camel milk and camel milk powder regarding willingness to pay.

## Conclusion

Although most preferred fresh milk, the mean rating of powder was still high. Powdered milk might be a good alternative in settings where a cold chain is difficult to maintain and shelf life is an issue. Factors such as place of residence and sex of pastoralist significantly influence preference of powdered milk. There is no significant difference in willingness to pay between fresh camel milk and powdered camel milk. Focusing on nutritional advantages, safety, and convenience regarding camel milk powder is crucial for rural and male pastoralists' adoption. Promotional efforts should be improved in accessibility and practicality without reducing cost.

## Introduction

Camel milk is a vital food source that supplies energy and nutrients to rural communities living in the arid regions of Africa and the Middle East [1,2]. Camel milk is a vital food source that supplies energy and nutrients to rural communities. Additionally, camel milk improves livelihoods and supports economic development for communities living in the arid regions of Africa and the Middle East, both domestically and internationally [3]. Pastoralists in subsistence farming systems keep camels mostly for their milk. They are famous for continuing to produce milk in the face of drought. When milk from cows, lambs, and goats is in short supply during dry seasons and drought years, camels are a very dependable source of milk [4,5].

According to statistics on livestock, there are approximately 35.5 million camels worldwide, with the largest populations found in Somalia, Sudan, Niger, Kenya, Chad, Ethiopia, Mali, Mauritania, and Pakistan [3]. In Ethiopia alone, there are about 2.4 million camels, primarily located in the southern and eastern pastoral regions of the country [6,7].

Even though over 80% of the world's camel population is found in Africa, with 60% residing in Eastern African countries, traditional camel milk producers create and consume the product locally, with the production process typically low-tech; this has subsequently created minimal global trade in camel milk [3]. Recent research found that camel milk, known for its genuine or alleged "medicinal" benefits, is moving beyond the margins, with global camel milk production having seen significant annual growth, surpassing 8% from 2009 to 2019, reflecting the increasing interest in this product [1,8].

Conversely, the pronounced seasonality in pastoral environments significantly affects the camel milk subsector. Although camel physiology enables milk production even during dry and drought periods, the overall milk supply tends to decrease. Additionally, during extended dry seasons, herders must travel long distances in search of feed and water [9]. The food industry uses milk powders for many purposes due to their ease of product development, processing, storage, and transportation. Additionally, milk powders are among the best methods for prolonging the shelf life of milk without drastically lowering its quality and nutrients [10,11].

The advancement of powdered milk production, which is the most effective way to preserve this highly perishable product for later use, has made it possible to include camel milk in the global dairy market. Moreover, since camel milk is often produced in remote areas far from where it is consumed, the only practical method for transporting it in large quantities is by removing its water content, which makes up 88% to 90% of its total weight [12].

Approximately 75,000 tons of camel milk are produced annually in Ethiopia; during the dry seasons, camels continue to produce milk for long periods despite the shortage of pasture, and most of this milk is consumed raw, while a smaller amount is taken in a fermented form [13]. For pastoralists in Ethiopia's Somali region, milk produced by a lactating camel not only nourishes the calf but also serves as an important nutritional source for people, helping to alleviate food shortages [2,14].

Although several studies have examined the nutritional benefits and technological processing of camel milk, there is a lack of empirical research on the sensory acceptability of camel milk in powdered form, especially in pastoralist communities like those in the Somali Region of Ethiopia. Conducting a targeted sensory trial is crucial to address this gap and guide the viability and design of future milk powder initiatives in the region. Therefore, this is aimed to investigating the sensory preference of camel milk powder compared to fresh camel milk among pastoralist communities in the Somali Region of Ethiopia.

## Method and materials

### Study area and period

The study was conducted in the Fafan and Shebelle zones in the Somali region between February and March 2025. Jigjiga is the capital of the Somali regional state in Ethiopia, situated 634 kilometers from the country's capital, Addis Ababa. As one of the six city council administrations in the region, Jigjiga operates under its own city administration and is divided into 30 kebeles; among these, 20 are classified as urban and 10 as rural. The projected population of the city is approximately 182,922 (CSA.2024). Shebelle (Gode and Adadle) is located 1183 kilometers from Addis Ababa and 549 kilometers away from Jigjiga.

### Study design

A single-blinded sensory crossover design was employed to evaluate the sensory preference of camel milk powder among pastoralist communities in the Somali Region. Participants were unaware of whether they were tasting fresh camel milk or camel milk powder.

### Source and study population

The source population for this study was all pastoralist communities in the Fafan and Shebelle Zones of the Somali region. The study population included randomly selected pastoralist communities from the Fafan and Shebelle Zones.

### Inclusion and exclusion criteria

The study included pastoralist communities in randomly selected zones. All pastoralist communities who are seriously ill, cannot respond, are absent, or refuse to participate in the study were excluded from the study.

## Sample size determination

The required sample size of the study participants was determined by using single population proportion formula.

$$n = \frac{\left(Z\frac{\alpha}{2}\right) 2 \times P\left(1-P\right)}{d^2}$$

With following assumptions;
n – The minimum sample size required
P – Estimated proportion of acceptance of camel milk powder among pastoralist communities.
d – Margin of error 5%(0.05)
Zα/2- Standard normal value at (1-ɑ) 95% confidence level
Prevalence (P) of acceptance powder camel milk was taken from a study conducted in Kenya, which was 19% [15].
Hence,

$$N = \frac{[Z\alpha2]^2 p(1\text{-}p) = (1.96)^2(0.19)(1-0.19)}{d^2 \quad (0.05^2)} = 236$$

The final sample, including a 5% non-response rate, was 248.

## Sampling technique and procedure

We purposefully selected two zones from the Somali region, ensuring that each zone included both urban and rural pastoralist communities. To construct the sampling frame, a list of pastoralist households was first obtained from local administrative offices in the selected kebeles within the Fafan and Shebelle Zones. Based on this household listing, the total sample size (n = 248) is proportionally allocated across kebeles according to population size and households. A systematic random sampling was used to select study subjects. The sampling interval (k) was determined by dividing the total number of households by the allocated sample size of the participants for that kebele, and random sampling was used to select the first household. If the selected household was absent or not eligible, the next immediate household was selected, and only one eligible adult was interviewed (**Fig 1**).

## Data collection method

Data was collected using an interviewer-administered, structured questionnaire designed to capture demographic information and participant responses for each milk sample. Four data collectors and two supervisors conducted the survey. The questionnaire was translated into Somali and back-translated into English for accuracy. Study participants received brief guidance on the Likert scale, but no sensory training was provided to prevent bias. The sequence was randomized, assigning either fresh/camel milk powder or powdered/fresh camel milk to each participant. Each participant was first given 20 ml of camel milk (either fresh or camel milk powder) and asked whether they preferred it, responding with either "yes" or "no." They were then provided with water to rinse their mouths before receiving a second 20 ml sample of the other type of camel milk. The acceptability of the second sample was recorded in the same manner.

## Study variables

Preference for powdered camel milk was the dependent variable, while place of residence, sex of pastoralist, age of pastoralist, milk expenditure, and milk liking level were the independent variables.

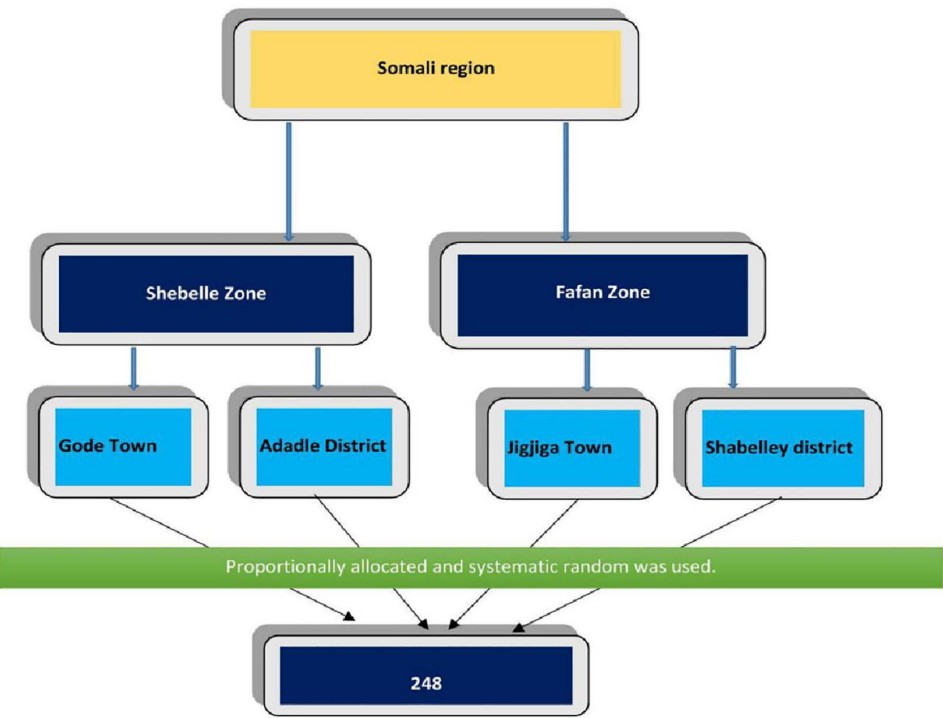

**Fig 1. The flow of the pastoralist communities' selection among residents in Somali region, Eastern Ethiopia.**

## Operational definition

**Camel milk powder Preference** was defined as the participant's positive response ("yes") to the question of whether they found the camel milk powder sample acceptable for consumption during a blinded taste test [16].

**Milk liking level** was assessed using a 5-point Likert scale (1 = Strongly dislike to 5 = Strongly like), following sensory evaluation standards described by Stone and Sidel [17].

**Milk spending** was assessed by asking participants their willingness to pay for camel milk using a four-point scale, adapted from consumer valuation methods used in food product studies [18].

## Data quality control

The quality of data was assured by proper designing and pre-testing (5%) of the questionnaires in those pastoralist communities that were not participating in the study with similar socio-demographic characteristics in order to ensure their validity. Training was given for both data collectors and supervisors by the principal investigator for one day. The sensory trial was single-blinded; participants did not know whether they tasted fresh or camel milk powder. Both fresh and camel milk powder samples were prepared using standard procedures (e.g., consistent dilution ratio, serving temperature, and volume: 20 ml per sample). Participants were offered water between samples to cleanse their taste. The training included discussion on the objectives of the study and on the contents of the questionnaire one by one, on procedures, on data collection techniques, and on the issues of the confidentiality of the responses. The collected data was checked by supervisors on a daily basis for any incompleteness, inconsistency, and possible corrections that were made each day based on the identified gaps accordingly. Inconsistencies and possible corrections were made each day based on the identified gaps accordingly.

## Data analysis and processing

Data were cleaned, coded, checked for completeness, and entered into Kobo Toolbox, then exported to STATA statistical software version 18 for analysis. Descriptive statistics such as mean, median, frequency, and percentage were used and presented in charts and tables. Bivariate analysis was conducted using binary logistic regression to assess the relative importance of exposure variables in relation to the outcome variable, expressed using odds ratios, and variables with p-values < 0.2 were included in the multivariable logistic regression analysis. In multivariable binary logistic regression, exposure variables with a p-value less than 0.05 and a 95% confidence interval were considered significantly associated with preference camel milk powder. An independent sample t-test was conducted to assess the mean differences in milk spending between the two types of camel milk (powdered vs. fresh). Additionally, the t-test was used to compare the mean milk liking levels between participants who tasted camel milk powder and those who tasted fresh camel milk.

## Ethical consideration

Ethical was reviewed and approved by the Jigjiga University Research Ethics Review Committee (Approval No. JJU-RERC/002/2025). An official permission letter was also submitted to the relevant administrative offices in the Shebelle and Fafan Zones. Informed, voluntary, written, and signed consent was obtained from the pastoralist communities after the purpose and potential benefits of the study were clearly explained. To ensure participant confidentiality, no identifying information was collected in the questionnaires. Participants were also informed that all data collected would be used solely for research purposes.

## Results

### Socio-demographic characteristics among pastoralist communities in Somali region, Eastern Ethiopia

A total of 248 pastoralists were selected for the study, of whom 242 participated, resulting in a response rate of 97.6%. The overall mean age of the participants was 34.9 years, with a standard deviation of SD ± 11.4 years. Among male participants, the mean age was 35.9 years (SD ± 11.5), while for female participants, it was 33.0 years (SD ± 11.1). Two-thirds of the participants, 159 (65.7%), were male. Additionally, more than half of the participants, 132 (54.6%), identified as rural pastoralists, and approximately half, 108 (44.6%), were aged between 30 and 45 years (Table 1).

### Participants' preference for the fresh or reconstructed milk in Somali Region

In the Somali Region, 167 pastoralists (69%; 95% CI: 62.9%–74.5%) preferred fresh camel milk, while 75 (31%; 95% CI: 25.5%–37.1%) preferred camel milk powder.

Table 1. Socio-demographic characteristics among pastoralist communities in Somali region, Eastern Ethiopia.

| Socio-demographic variables | Categories | Frequency | Percent |
|---|---|---|---|
| Place of residence | Rural | 106 | 43.8 |
| | Urban | 136 | 56.2 |
| Sex of participant | Male | 154 | 63.6 |
| | Female | 88 | 36.4 |
| Age group | <30 years | 92 | 38.0 |
| | 30 - 45 years | 108 | 44.6 |
| | >45 years | 42 | 17.4 |

### Independent samples T-Test for comparing milk liking levels based on milk type preference (powdered vs. fresh camel milk)

The mean and standard deviation of liking level for fresh camel milk was 4.1 (±0.78). The 95% confidence interval indicates that we can be 95% confident that the true mean liking level for fresh camel milk lies between 4.0 and 4.2. The mean and standard deviation of liking level for camel milk powder was 3.6 (±0.94). The confidence interval suggests that the true mean liking level for camel milk powder is likely between 3.4 and 3.8. Pastoralists who preferred fresh camel milk had significantly higher milk liking scores (mean = 4.1) compared to those who preferred camel milk powder (mean = 3.6). The difference of 0.5 points is statistically significant ($p < 0.001$), and the 95% confidence interval does not include zero, indicating a true difference in liking levels between the fresh camel milk and camel milk powder (Table 2).

### Independent samples T-Test for comparing willingness to pay based on milk type preference (powdered vs. Fresh camel milk)

The mean and standard deviation of willingness to pay for fresh camel milk were 3.0 (SD ± 0.65) and 3 (95% CI, 2.9 and 3.1). The mean and standard deviation of willingness to pay for camel milk powder was around 2.9 (SD ± 0.73) and 2.9 (95% CI, 2.8 and 3.1). Although pastoralists who preferred fresh camel milk showed a slightly higher mean willingness to pay (3.0) compared to those who preferred camel milk powder (2.9), this difference was not statistically significant (*p-value* = 0.3136) (Table 3).

### Bi-variable and multivariable analysis to preference powder of camel milk

Both bi-variable and multivariable logistic regression analyses were done. Variables included in the analysis were place of residence, sex, and age. First, a bivariate analysis was done, and variables with a cut-off point of <0.2 in the bi-variable analysis were taken into the multivariable analysis, and finally, the multivariate analysis of variables less than 0.05 were declared significant. In the multivariable logistic regression analysis, two variables were found to be significantly associated with preference camel milk powder.

The odds of preferred camel milk powder were 2.2 times higher among urban pastoralists compared to rural pastoralists (AOR = 2.2; 95% CI: 1.2–4.3) with (*p-value* of 0.015).

**Table 2. Independent Samples T-Test for Comparing Milk Liking Levels Based on Milk Type Preference (Powdered vs. Fresh Camel Milk) in Somali Region. Ethiopia.**

| Group | Observation | Mean | Standard error | Standard deviation | 95% Confidence interval |
|---|---|---|---|---|---|
| Fresh camel Milk | 167 | 4.1 | 0.06 | 0.78 | (4.0, 4.2) |
| Powder camel | 75 | 3.6 | 0.11 | 0.94 | (3.4, 3.8) |
| Combined | 242 | 4.0 | 0.06 | 0.86 | (3.8, 4.1) |
| Difference | | 0.5 | 0.12 | | (0.3, 0.8) |

**Table 3. Independent Samples T-Test for Comparing Willingness to Pay Based on Milk Type Preference (Powdered vs. Fresh Camel Milk) in Somali Region, Ethiopia.**

| Group | Observation | Mean | Standard error | Standard deviation | 95% Confidence interval |
|---|---|---|---|---|---|
| Fresh camel Milk | 167 | 3.0 | 0.05 | 0.65 | (2.9, 3.1) |
| Powder camel | 75 | 2.9 | 0.09 | 0.73 | (2.8, 3.1) |
| Combined | 242 | 3.0 | 0.04 | 0.68 | (2.9, 3.1) |
| Difference | | 0.1 | 0.09 | | (−0.1, 0.3) |

Similarly, female pastoralists had twice the odds of Preferred camel milk powder compared to male pastoralists (AOR = 2.0; 95% CI: 1.1–3.7) with (*p-value* of 0.024). (**Table 4**).

Rural pastoralists showed a strong preference for milk, with 58 participants (54.7%) indicating that they "liked" it, while 58 participants (42.7%) from the urban pastoralist also expressed a positive sentiment. Female participants were particularly motivated toward milk, with 44 (50.0%) stating that they "liked" it, and 72 male participants (46.8%) shared the same view. Among age categories, participants aged 30–45 displayed a preference, with 53 (49.1%) stating that they "liked" the milk, while those under 30 showed a neutral response, with 50 participants (54.4%) expressing similar feelings. Fresh camel milk was particularly favored, with 82 participants (49.1%) indicating that they "liked" it, while 34 participants (45.3%) preferred camel milk powder (**Fig 2**).

In terms of place of residence, the rural pastoralist showed a notable preference, with 65 (61.3%) participants indicating they were willing to pay a fair amount for milk, slightly lower than the urban pastoralist at 90 (66.2%). Regarding sex, female pastoralist demonstrated a stronger willingness, with 60 (68.2%) participants expressing their readiness to pay a fair amount, while 95 (61.7%) male participants reported the same. Among the age categories, those aged under 30 included 62 (67.4%) participants who were willing to pay a fair amount. Regarding milk type, fresh camel milk was favored, with 108 (64.7%) participants indicating they were willing to pay a fair amount, compared to 47 (62.7%) participants for camel milk powder (**Fig 3**).

## Discussion

This finding showed that a significantly larger proportion of pastoralists in the Somali Region prefer fresh camel milk compared to camel milk powder. In this study, the preference of camel milk powder among pastoralist communities in the Somali Region was 75 (31%) with the CI of (25.46%, 37.13%). Our findings are similar to a study conducted in Nigeria [19]. Preference of camel milk powder in both Ethiopia and Nigeria is influenced by their shared pastoralist traditions and dependence on livestock for livelihoods, and large segments of the population in both countries are accustomed to consuming fresh milk, which contributes to only moderate preference for powdered alternatives. Additionally, they encounter comparable obstacles in promoting powdered milk, including low awareness of its nutritional value and differing views on product quality. Despite this, the relatively high score for powdered camel milk may reflect its practical advantages, such as longer shelf life, consistent taste, and ease of storage, particularly in hot climates, and these benefits are likely better known by urban residents and women, who mostly manage household food choices.

On the other hand, the preference of camel milk powder in this study was lower than in a study conducted in Kenya [20]. This difference may be attributed to cultural norms, economic conditions, and more advanced market development in Kenya. A better developed dairy sector and increased public awareness of powdered milk, bolstered by effective

**Table 4. Bivariate and multivariate analyses of factors associated with preference of camel milk powder in the Somali Region.**

| Variables | Category | Preference camel Milk powder | | Bi-variable analysis | | Multi-variable analysis | |
|---|---|---|---|---|---|---|---|
| | | Yes N (%) | No N (%) | COR (95% CI) | P-value | AOR (95% CI) | P-value |
| Place Residence | Rural | 21(19.8) | 85(80.2) | 1 | 0.001 | 1 | 0.015* |
| | Urban | 54(39.7) | 82(60.3) | 2.7(1.5, 4.1) | | 2.2(1.2, 4.3) | |
| Sex | Male | 36(23.4) | 118(76.6) | 1 | 0.001 | 1 | |
| | Female | 39(44.3) | 49(55.7) | 2.6(1.6, 4.6) | | 2.0(1.1, 3.7) | 0.024* |
| Age category | <30 years | 30(32.6) | 62(67.4) | 1.2(0.5, 2.7) | 0.640 | 1.1(0.5, 2.7) | 0.682 |
| | 30–45 years | 33(30.6) | 75(69.4) | 1.1(0.5, 2.4) | 0.812 | 1.3(0.6, 3.0) | 0.540 |
| | >45 years | 12(28.6) | 30(71.4) | 1 | | 1 | |

NB *= Significant, CI confidence interval, COR = crude odds ratio, AOR = adjusted odds ratio

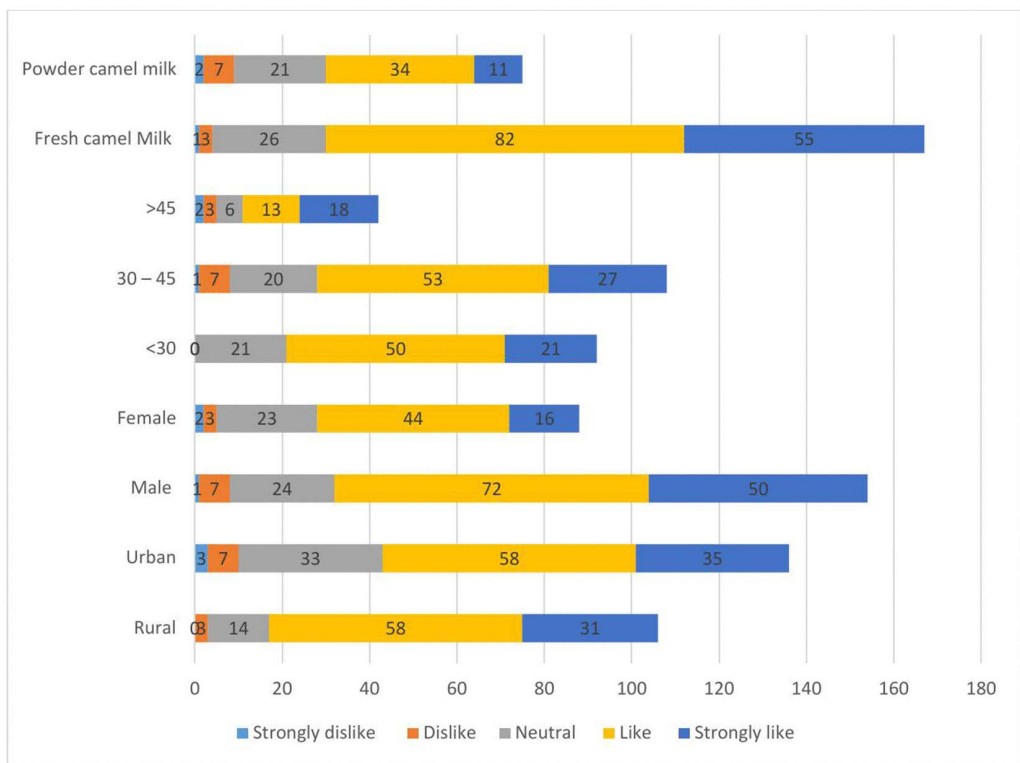

**Fig 2. Level of milk liking among pastoralist communities in Somali region, Eastern Ethiopia.**

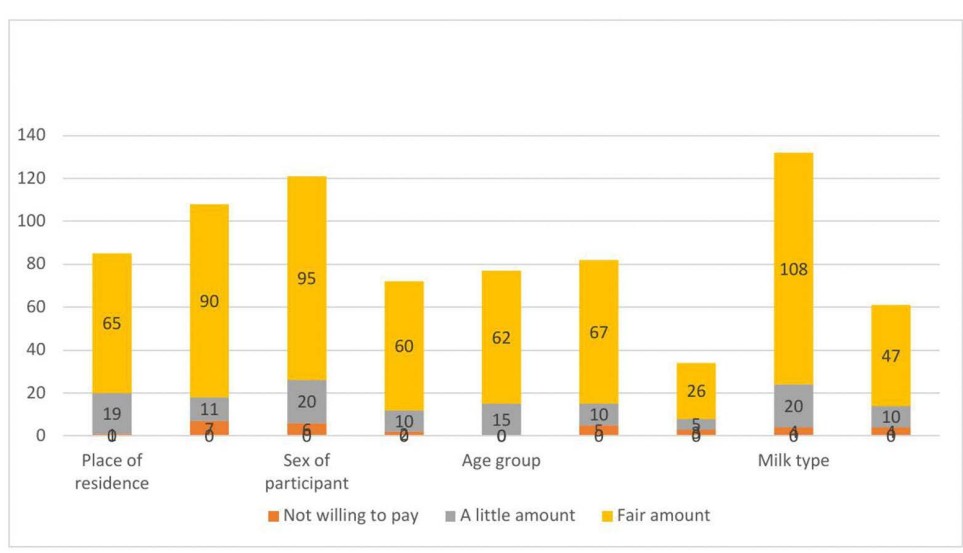

**Fig 3. Willingness to pay for camel milk among pastoralist communities in Somali Region, Eastern Ethiopia.**

marketing and consumer education initiatives, are advantageous for Kenya. Its popularity is hindered in the Somali Region by inadequate infrastructure, a cultural preference for fresh milk, and a lack of knowledge about the advantages of powdered milk.

Place of residence is significantly associated with preference of camel milk powder. Urban pastoralists were 2.2 times more likely to accept camel milk powder compared to rural pastoralists (AOR = 2.2; 95% CI: 1.17–4.28). This study consisted of the study conducted, Comparative Analysis of Acceptability of Camel Milk among Rural and Urban Consumers in Kenya [21]. Urban populations tend to have better access to information and resources, which helps people become more aware of the nutritional advantages of powdered milk. The discrepancy may be due to the broader availability of powdered milk products in urban markets, where effective marketing strategies and better consumer exposure encourage their preference.

Sex of the pastoralist was significantly associated with preference of camel milk powder. Female pastoralists had twice the odds of accepting camel milk powder compared to male pastoralists (AOR = 2.0; 95% CI: 1.09–3.66). This possible may be because women are more likely to manage the food resources in the home and are more willing to provide their families with a variety of dietary options. Comparatively speaking to males, women are more likely to accept camel milk powder because they are more proactive in embracing innovations that can improve household welfare.

This study showed that the liking levels between the fresh camel milk and camel milk powder are not the same. This finding is consistent with the study conducted in European and Mediterranean countries [22]. The taste and apparent freshness of fresh camel milk make it the preferred option above its powdered equivalent. Associating fresh camel milk with superior flavor, participants in sensory evaluations frequently expressed a stronger preference for it.

There is no significant difference in the mean willingness to pay between fresh camel milk and powdered camel milk; this difference was not statistically significant (p = 0.3136). This finding is in line with the study conducted in Aba'ala woreda, Afar Region [23]. This might be because other elements, such as taste preference, familiarity, accessibility, or perceptions of quality, can have a greater impact on consumer choice than price. Given that customers seem equally willing to pay for either product when price is the only consideration, interventions meant to increase the adoption of camel milk powder should concentrate on addressing these non-price aspects. These results indicate encouraging opportunities for future camel milk powder initiatives and targeted promotion to urban and female consumers, investment in local production, and integration into nutrition programs, especially where electricity for cold chains is limited, which could enhance both market reach and public health outcomes.

### Limitation of the study

The sensory evaluation was established on only one exposure, which may not show what people like over time longitudinally. Socio-cultural biases towards fresh milk could also have influenced participant responses despite using a single-blinded study design

### Conclusion

Although the majority preferred fresh camel milk, the average rating for powdered milk remained high. This suggests that powdered milk could serve as a suitable alternative in settings where maintaining a cold chain is challenging and shelf life is a concern. Place of residence and sex of pastoralist are significantly associated with preference of powder camel milk. While participants showed a significant preference for fresh camel milk over camel milk powder in terms of liking, the difference in their willingness to pay for the two types was not statistically significant. Implementing focused educational efforts that highlight the product's nutritional advantages for men is crucial. The nutritional advantages, safety, and convenience of camel milk powder should be highlighted in focused awareness and education initiatives to increase rural

pastoralists' adoption of the camel milk powder. Promotional efforts for camel milk powder should concentrate on increasing its accessibility and practicality without reducing its cost.

## Supporting information

**S1 File. Data used in this manuscript.**
(XLS)

**S2 File. ANNEX.**
(DOCX)

**S3 File. Inclusivity-in-global-research-questionnaire.**
(DOCX)

## Author contributions

**Conceptualization:** Ahmed Mohammed Ibrahim, Seid Mohammed Ali, Yahya Maidane Osman, Fathiya Budul Ismail, Mohamed Omar Osman, Mukhtar Harir Hussein, Abdifatah Muktar Muhummed, Ramadan Budul Yusuf, Raymond Place, Jan Hattendorf, Pauline Rouchon, Rea Tschopp, Pascal Mäser, Jakob Zinsstag.

**Data curation:** Ahmed Mohammed Ibrahim, Seid Mohammed Ali, Yahya Maidane Osman, Fathiya Budul Ismail, Mohamed Omar Osman, Mukhtar Harir Hussein, Abdifatah Muktar Muhummed, Ramadan Budul Yusuf, Raymond Place, Jan Hattendorf, Pauline Rouchon, Rea Tschopp, Pascal Mäser, Jakob Zinsstag.

**Formal analysis:** Ahmed Mohammed Ibrahim, Seid Mohammed Ali, Yahya Maidane Osman, Fathiya Budul Ismail, Mohamed Omar Osman, Mukhtar Harir Hussein, Abdifatah Muktar Muhummed, Ramadan Budul Yusuf, Raymond Place, Jan Hattendorf, Pauline Rouchon, Rea Tschopp, Pascal Mäser, Jakob Zinsstag.

**Funding acquisition:** Mohamed Omar Osman.

**Investigation:** Seid Mohammed Ali, Yahya Maidane Osman, Fathiya Budul Ismail, Mohamed Omar Osman, Mukhtar Harir Hussein, Abdifatah Muktar Muhummed, Ramadan Budul Yusuf, Raymond Place, Jan Hattendorf, Pauline Rouchon, Rea Tschopp, Pascal Mäser, Jakob Zinsstag.

**Methodology:** Ahmed Mohammed Ibrahim, Seid Mohammed Ali, Yahya Maidane Osman, Fathiya Budul Ismail, Mohamed Omar Osman, Mukhtar Harir Hussein, Abdifatah Muktar Muhummed, Ramadan Budul Yusuf, Raymond Place, Jan Hattendorf, Pauline Rouchon, Rea Tschopp, Pascal Mäser, Jakob Zinsstag.

**Project administration:** Ahmed Mohammed Ibrahim, Seid Mohammed Ali, Yahya Maidane Osman, Fathiya Budul Ismail, Mohamed Omar Osman, Mukhtar Harir Hussein, Abdifatah Muktar Muhummed, Ramadan Budul Yusuf, Raymond Place, Jan Hattendorf, Pauline Rouchon, Rea Tschopp, Jakob Zinsstag.

**Resources:** Yahya Maidane Osman, Mohamed Omar Osman, Mukhtar Harir Hussein, Abdifatah Muktar Muhummed, Ramadan Budul Yusuf, Raymond Place, Jan Hattendorf, Rea Tschopp, Jakob Zinsstag.

**Software:** Ahmed Mohammed Ibrahim, Seid Mohammed Ali, Yahya Maidane Osman, Fathiya Budul Ismail, Mohamed Omar Osman, Mukhtar Harir Hussein, Abdifatah Muktar Muhummed, Ramadan Budul Yusuf, Raymond Place, Jan Hattendorf, Pauline Rouchon, Rea Tschopp, Pascal Mäser, Jakob Zinsstag.

**Supervision:** Ahmed Mohammed Ibrahim, Seid Mohammed Ali, Yahya Maidane Osman, Fathiya Budul Ismail, Mohamed Omar Osman, Mukhtar Harir Hussein, Abdifatah Muktar Muhummed, Ramadan Budul Yusuf, Raymond Place, Jan Hattendorf, Pauline Rouchon, Rea Tschopp, Jakob Zinsstag.

**Validation:** Ahmed Mohammed Ibrahim, Seid Mohammed Ali, Yahya Maidane Osman, Fathiya Budul Ismail, Mohamed Omar Osman, Mukhtar Harir Hussein, Abdifatah Muktar Muhummed, Ramadan Budul Yusuf, Raymond Place, Jan Hattendorf, Pauline Rouchon, Rea Tschopp, Pascal Mäser, Jakob Zinsstag.

**Visualization:** Ahmed Mohammed Ibrahim, Seid Mohammed Ali, Yahya Maidane Osman, Fathiya Budul Ismail, Mohamed Omar Osman, Mukhtar Harir Hussein, Abdifatah Muktar Muhummed, Ramadan Budul Yusuf, Raymond Place, Jan Hattendorf, Pauline Rouchon, Rea Tschopp, Pascal Mäser, Jakob Zinsstag.

**Writing – original draft:** Ahmed Mohammed Ibrahim, Seid Mohammed Ali, Yahya Maidane Osman, Fathiya Budul Ismail, Mohamed Omar Osman, Mukhtar Harir Hussein, Abdifatah Muktar Muhummed, Ramadan Budul Yusuf, Raymond Place, Jan Hattendorf, Pauline Rouchon, Rea Tschopp, Pascal Mäser, Jakob Zinsstag.

**Writing – review & editing:** Ahmed Mohammed Ibrahim, Seid Mohammed Ali, Yahya Maidane Osman, Fathiya Budul Ismail, Mohamed Omar Osman, Mukhtar Harir Hussein, Abdifatah Muktar Muhummed, Ramadan Budul Yusuf, Raymond Place, Jan Hattendorf, Pauline Rouchon, Rea Tschopp, Pascal Mäser, Jakob Zinsstag.

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
