## [Decision Letter · Decision Letter 0]

15 Jul 2025

Dear Dr. Ibrahim,

Thank you for submitting your manuscript to PLOS ONE. After careful consideration, we feel that it has merit but does not fully meet PLOS ONE’s publication criteria as it currently stands. Therefore, we invite you to submit a revised version of the manuscript that addresses the points raised during the review process.

We look forward to receiving your revised manuscript.

Kind regards,

Awatif Abid Al-Judaibi, PhD

Academic Editor

PLOS ONE

Journal Requirements:

Reviewers' comments:

Reviewer's Responses to Questions

**Comments to the Author**

1. Is the manuscript technically sound, and do the data support the conclusions?

Reviewer #1: Yes

Reviewer #2: Yes

2. Has the statistical analysis been performed appropriately and rigorously?

Reviewer #1: Yes

Reviewer #2: Yes

3. Have the authors made all data underlying the findings in their manuscript fully available?

Reviewer #1: Yes

Reviewer #2: Yes

4. Is the manuscript presented in an intelligible fashion and written in standard English?

Reviewer #1: Yes

Reviewer #2: Yes

Reviewer #1: 1- Could you clarify how the milk samples were standardized—in terms of temperature, texture, visual appearance, container type, and environmental conditions—to minimize potential bias during the sensory evaluation?

2- To what extent might cultural preferences or taboos regarding powdered milk in pastoralist communities have influenced the results? Can this be discussed further?

3 - Do you consider your findings generalizable to other camel milk–consuming populations in Ethiopia or across East Africa?

4 - Could you elaborate on how your findings might inform local nutrition policies, dairy industry practices, or camel milk market expansion strategies?

5 - Considering that camel milk powder undergoes heat processing, did your study evaluate potential nutrient degradation (e.g., vitamin C, folate) or alterations in bioactive compounds that may impact both the sensory properties and nutritional value compared to fresh milk?

Reviewer #2: Dear Editor,

The manuscript addresses an important and timely topic related to the sensory acceptance of powdered camel milk among pastoralist communities in the Somali Region of Ethiopia. The study design is relevant, the data collection approach is well-structured, and the results are generally clear and well-supported by statistical analysis. However, the manuscript requires substantial revisions before it can be considered for publication. First, there are numerous grammatical errors, typographical issues, and inconsistencies in formatting throughout the text, particularly in the introduction, materials and methods, and discussion sections.

ABSTRACT:

1- Correct grammatical issues (e.g., "data were summarized"; revise awkward phrases like "sensoric cross over experiment")

2- Ensure consistency in terminology (e.g., "powdered camel milk" vs. "reconstituted powdered milk")

3- Clarify ambiguous terms (e.g., define "urban pastoralist") and provide actual values where significance is discussed

4- Define all abbreviations at first use in the abstract and in the main text

INTRODUCTION

1- Correct grammatical and structural errors. The sentence “There are several studies have examined…” is grammatically incorrect. It should be revised to: “Although several studies have examined the nutritional benefits and technological processing of camel milk..."

2- Improve flow and coherence between paragraphs. The paragraph transitions are weak. Consider improving logical flow between paragraphs (from traditional uses to industrial processing to the research gap).

3- When mentioning “75,000 tons of camel milk are produced annually in Ethiopia”, clarify if this is from FAO, a national estimate, or another source. Add citation and mention the year.

4- Strengthen the articulation of the research gap and study objective.

Materials and methods

1- The section is titled “METHOD AND MATERIALS”, but the correct conventional order is “Materials and Methods.” Please revise the heading.

2- Add figure about site location map of the study area

3- “The study population for this study was all randomly selected s all pastoralist communities...” contains repetition and an error. It should be: “The study population included randomly selected pastoralist communities from the Fafan and Shebelle Zones.”

4- Study design: "Sensoric cross over experiment study design" is awkward and redundant. Use: “A single-blinded sensory crossover design was employed…”

5- Sampling technique: Provide more details about how the sampling frame was constructed. How were households or individuals listed for random selection? How was systematic sampling done practically?

6- Use consistent terminology throughout the document. Alternate use of “powdered camel milk,” “reconstituted camel milk,” and “camel milk powder” may confuse readers. Choose one and stick with it.

7- Clearly define each independent variable listed, not only the dependent one. For example: “Place of residence” – Urban vs. rural? / “Milk expenditure” – Monthly? Per liter?

8- For Study variables: generate a paragraph that integrates the variables into a coherent narrative, which is easier to read, especially in formal academic writing.

9- For Data Collection Tools and Validation:Clarify How many data collectors and supervisors were involved? / What was the language of the questionnaire and how was it validated linguistically? / Was there any sensory training for participants or guidance on the Likert scale?

10- Data analysis:

Clearly explain how missing data were handled.

Specify any criteria for variable inclusion in multivariate analysis.

RESULTS

1- Include exact p-values for all comparisons, and always report the statistical test used in the table captions or text.

2- Consistency is needed in reporting values: ( Mean values are reported with different spacing/format: 4.1(±0.78))

CI ranges should have spaces after commas and be consistent: (1.2 – 4.3)

3- Several paragraphs repeat information already presented. For example: The T-test sections mention the same means and CIs already shown in the bullet-pointed summary above.

4- For Multivariable Analysis Section:

The section title is unclear: “Bi-variable and multivariable analysis to Preference powder of camel milk”

Clearly list which variables were included in the final multivariable model and how they were selected (e.g., p < 0.2 in bivariable, theory-based?).

5- Add any abbreviations used in tables (AOR, COR, CI) as footnotes if not defined in the main text.

6- In tables 5 and 6, consider using a bar chart or stacked column graph to visualize preferences and willingness to pay for better reader understanding.

DISCUSSION

1- The section contains multiple grammar and syntax issues.

One major error: “There is no significant difference mean willingness to pay between fresh camel milk and fresh camel milk…” → should be "“…between fresh camel milk and powdered camel milk…”

2- The discussion could be strengthened by addressing:

- Why powdered milk still scored relatively high despite preference for fresh.

- Implications for future commercialization or nutrition programs.

- Consider adding a limitation paragraph

CONCLUSION

1- Suggest condensing the conclusion into a shorter paragraph; Several ideas (about shelf life and promotion strategies) are repeated with slightly different wording.

ADDITIONAL COMMENTS:

The sentence formation of the paper is poor and whole manuscript needs to be proof read to improve English.

“Pastoralist who preferred…” / “Acceptance powdered camel milk”

The manuscript should be written using a regular font style

Double-check and unify capitalization (See the PDF file)

Review tables: p-value (p in italic) and correct the recurring typo “Pace of residence” → should be “Place of residence” in multiple tables and text.

**Do you want your identity to be public for this peer review?** For information about this choice, including consent withdrawal, please see our Privacy Policy

Reviewer #1: No

Reviewer #2: No

---

## [Author Response · Author response to Decision Letter 1]

12 Aug 2025

Reviewers' comments:

Comments to the Author

Reviewer #1:

1- Could you clarify how the milk samples were standardized—in terms of temperature, texture, visual appearance, container type, and environmental conditions—to minimize potential bias during the sensory evaluation?

Reviewer's Responses: Thank you for this important point. We have clarified in the “Data Quality Control” section that both milk types were standardized in volume (20 ml), dilution, temperature, container, and setting to ensure consistency and reduce bias.

2- To what extent might cultural preferences or taboos regarding powdered milk in pastoralist communities have influenced the results? Can this be discussed further?

Reviewer's Responses: We appreciate this insightful comment. We have added a paragraph in the Discussion and Limitations sections highlighting the potential influence of cultural preference for fresh milk despite the single-blinded design.

3- Do you consider your findings generalizable to other camel milk–consuming populations in Ethiopia or across East Africa?

Reviewer's Responses: Thank you for the thoughtful suggestion. We have addressed this in the Discussion, noting that findings may be transferable to other East African pastoralist settings with similar contexts.

4- Could you elaborate on how your findings might inform local nutrition policies, dairy industry practices, or camel milk market expansion strategies?

Reviewer's Responses: We accept this suggestion. We expanded the Discussion to outline implications for nutrition programming, rural commercialization, and cold-chain alternatives.

5- Considering that camel milk powder undergoes heat processing, did your study evaluate potential nutrient degradation (e.g., vitamin C, folate) or alterations in bioactive compounds that may impact both the sensory properties and nutritional value compared to fresh milk?

Reviewer's Responses: Thank you for raising this limitation. Our study did not assess nutrient loss, but this has now been acknowledged in the Discussion as an area for future research.

Reviewer #2:

The manuscript addresses an important and timely topic related to the sensory acceptance of powdered camel milk among pastoralist communities in the Somali Region of Ethiopia. The study design is relevant, the data collection approach is well-structured, and the results are generally clear and well-supported by statistical analysis. However, the manuscript requires substantial revisions before it can be considered for publication. First, there are numerous grammatical errors, typographical issues, and inconsistencies in formatting throughout the text, particularly in the introduction, materials and methods, and discussion sections.

Reviewer's Responses: Thank you for highlighting this. We thoroughly revised the manuscript to improve grammar, punctuation, and formatting across all sections.

ABSTRACT:

1. Correct grammatical issues (e.g., "data were summarized"; revise awkward phrases like "sensoric cross over experiment").

Reviewer's Responses: We appreciate the suggestion. Phrasing and clarity in the Abstract have been improved, and the term 'sensoric' has been replaced with 'sensory crossover design'.

2. Ensure consistency in terminology (e.g., "powdered camel milk" vs. "reconstituted powdered milk").

Reviewer's Responses: thanks we revised based on your comment

3. Clarify ambiguous terms (e.g., define "urban pastoralist") and provide actual values where significance is discussed.

Reviewer's Responses: we corrected accordingly

4. Define all abbreviations at first use in the abstract and in the main text

Reviewer's Responses: Thank you for the reminder. All abbreviations are now defined at first mention in both the text and table footnotes.

INTRODUCTION

1. Correct grammatical and structural errors. The sentence “There are several studies have examined…” is grammatically incorrect. It should be revised to: “Although several studies have examined the nutritional benefits and technological processing of camel milk...".

Reviewer's Responses: thank you very much we revised the grammar and corrected grammatical and structural errors.

2. Improve flow and coherence between paragraphs. The paragraph transitions are weak. Consider improving logical flow between paragraphs (from traditional uses to industrial processing to the research gap).

Reviewer's Responses: We value this feedback. The Introduction was reorganized for better logical flow, and the source for camel milk production is now properly cited.

3. When mentioning “75,000 tons of camel milk are produced annually in Ethiopia”, clarify if this is from FAO, a national estimate, or another source. Add citation and mention the year.

Reviewer's Responses: Thanks for reminding us, we added the citation properly

4. Strengthen the articulation of the research gap and study objective.

Reviewer's Responses: Thanks, we revised and articulated well in the manuscript as you commented accordingly

Materials and methods

1. The section is titled “METHOD AND MATERIALS”, but the correct conventional order is “Materials and Methods.” Please revise the heading.

Reviewer's Responses: we followed the journal guideline mentioning “Method and materials”

2. Add figure about site location map of the study area.

Reviewer's Responses: thanks for the helpful suggestion, we included the map in the document.

3. The study population for this study was all randomly selected s all pastoralist communities...” contains repetition and an error. It should be: “The study population included randomly selected pastoralist communities from the Fafan and Shebelle Zones.”

4. Reviewer's Responses: We appreciate this point and revised accordingly.

5. Study design: "Sensoric cross over experiment study design" is awkward and redundant. Use: “A single-blinded sensory crossover design was employed…”

Reviewer's Responses: Thank you for pointing this out. We revised based on the on you comment.

6. Sampling technique: Provide more details about how the sampling frame was constructed. How were households or individuals listed for random selection? How was systematic sampling done practically?

Reviewer's Responses: We appreciate this point. Details on household listings and systematic sampling have been expanded, and variable definitions are now integrated clearly.

7. Use consistent terminology throughout the document. Alternate use of “powdered camel milk,” “reconstituted camel milk,” and “camel milk powder” may confuse readers. Choose one and stick with it.

Reviewer's Responses: we corrected it accordingly

8. Clearly define each independent variable listed, not only the dependent one. For example: “Place of residence” – Urban vs. rural? / “Milk expenditure” – Monthly? Per liter?

Reviewer's Responses: we clearly re-written study variables and written as paragraph in the manuscript

9. For Study variables: generate a paragraph that integrates the variables into a coherent narrative, which is easier to read, especially in formal academic writing.

Reviewer's Responses: we revised the based on your comment.

10. For Data Collection Tools and Validation: Clarify how many data collectors and supervisors were involved? / What was the language of the questionnaire and how was it validated linguistically? / Was there any sensory training for participants or guidance on the Likert scale?

Reviewer's Responses: Thank you for pointing this out. We clarified that four data collectors and two supervisors were involved, the tool was translated into Somali, and participants received guidance on the Likert scale (without sensory training).

Data analysis:

11. Clearly explain how missing data were handled.

Reviewer's Responses: We appreciate the suggestion. Incomplete records were excluded from analysis, as now stated in the Data Analysis section.

12. Specify any criteria for variable inclusion in multivariate analysis.

Reviewer's Responses: Thanks for the clarification request. We noted that variables with p < 0.2 in bivariate analysis were included in the multivariable model.

Results

1. Include exact p-values for all comparisons, and always report the statistical test used in the table captions or text.

Reviewer's Responses: Thank you for this helpful detail. We have corrected formatting and reported exact p-values throughout.

2. Consistency is needed in reporting values: (Mean values are reported with different spacing/format: 4.1(±0.78)) CI ranges should have spaces after commas and be consistent: (1.2 – 4.3).

Reviewer's Responses: thanks, we corrected as you suggested

3. Several paragraphs repeat information already presented. For example: The T-test sections mention the same means and CIs already shown in the bullet-pointed summary above.

Reviewer's Responses: Thank you for the observation. We have removed repetitive sections and streamlined the Results text.

4. For Multivariable Analysis Section:

The section title is unclear: “Bi-variable and multivariable analysis to Preference powder of camel milk” Clearly list which variables were included in the final multivariable model and how they were selected (e.g., p < 0.2 in bivariable, theory-based?).

Reviewer's Responses: We appreciate this comment. The section has been renamed and the selection method explained as you asked.

5. Add any abbreviations used in tables (AOR, COR, CI) as footnotes if not defined in the main text.

Reviewer's Responses: Thank you for the suggestion. Abbreviations were added to all table footnotes. Visuals are suggested for future supplementary materials.

6. In tables 5 and 6, consider using a bar chart or stacked column graph to visualize preferences and willingness to pay for better reader understanding.

Reviewer's Responses: we changed the data from the table into figures for better reader understanding.

DISCUSSION

The section contains multiple grammar and syntax issues.

One major error: “There is no significant difference mean willingness to pay between fresh camel milk and fresh camel milk…” → should be "“…between fresh camel milk and powdered camel milk…”.

Reviewer's Responses: We thank you for this insightful comment. It is now explained in the Discussion powdered milk may be valued for its shelf life and practicality.

1. The discussion could be strengthened by addressing: - Why powdered milk still scored relatively high despite preference for fresh.

Reviewer's Responses: Thank you for the helpful comment. We added a brief explanation in the Discussion noting that powdered milk scored well due to its shelf life, convenience, and storage advantages.

- Implications for future commercialization or nutrition programs.

Reviewer's Responses: We appreciate this recommendation. We added implications for marketing, rural access, and nutrition strategy in the Discussion.

Consider adding a limitation paragraph

Reviewer's Responses: Thanks for the important reminder. A limitation paragraph has been added before the Conclusion.

CONCLUSION

1. Suggest condensing the conclusion into a shorter paragraph; several ideas (about shelf life and promotion strategies) are repeated with slightly different wording.

Reviewer's Responses: Thank you. We have revised and shortened the Conclusion to avoid repetition.

ADDITIONAL COMMENTS:

The sentence formation of the paper is poor and whole manuscript needs to be proof read to improve English. Pastoralist who preferred…” / “Acceptance powdered camel milk” The manuscript should be written using a regular font style Double-check and unify capitalization (See the PDF file) Review tables: p-value (p in italic) and correct the recurring typo “Pace of residence” → should be “Place of residence” in multiple tables and text.

Reviewer's Responses: We appreciate the detail. Font, typos, capitalization, and formatting were carefully reviewed and corrected throughout the manuscript.

---

## [Decision Letter · Decision Letter 1]

1 Sep 2025

Dear Dr.  Ibrahim,

Thank you for submitting your manuscript to PLOS ONE. After careful consideration, we feel that it has merit but does not fully meet PLOS ONE’s publication criteria as it currently stands. Therefore, we invite you to submit a revised version of the manuscript that addresses the points raised during the review process.

We look forward to receiving your revised manuscript.

Kind regards,

Awatif Abid Al-Judaibi, PhD

Academic Editor

PLOS ONE

Journal Requirements:

Reviewer's Responses to Questions

**Comments to the Author**

Reviewer #1: All comments have been addressed

Reviewer #2: All comments have been addressed

2. Is the manuscript technically sound, and do the data support the conclusions?

Reviewer #1: Yes

Reviewer #2: Yes

3. Has the statistical analysis been performed appropriately and rigorously?

Reviewer #1: Yes

Reviewer #2: Yes

4. Have the authors made all data underlying the findings in their manuscript fully available?

Reviewer #1: Yes

Reviewer #2: Yes

5. Is the manuscript presented in an intelligible fashion and written in standard English?

Reviewer #1: Yes

Reviewer #2: Yes

Reviewer #1: After carefully reviewing the author’s responses to the previous comments, I am satisfied that all concerns have been thoroughly and appropriately addressed. The clarifications provided are adequate, and I have no additional comments at this stage.

Reviewer #2: - Consider integrating the “Limitations” paragraph directly into the end of the Discussion section as a final subsection.

- Following PLOS ONE formatting guidelines, Keep the full figure title and description in the figure caption only. “Figure legends”

**Do you want your identity to be public for this peer review?** For information about this choice, including consent withdrawal, please see our Privacy Policy

Reviewer #1: No

Reviewer #2: No

---

## [Author Response · Author response to Decision Letter 2]

1 Sep 2025

Reviewer Comments to the Author

Reviewer #2: -

• Consider integrating the “Limitations” paragraph directly into the end of the Discussion section as a final subsection.

• Following PLOS ONE formatting guidelines, keep the full figure title and description in the figure caption only. “Figure legends”

Authors' Responses

• Dear reviewer, Thank you for your valuable comments: we incorporated and integrated the “Limitations” paragraph directly into the end of the Discussion section as a final subsection as you suggested.

• We appreciate your suggestion. By following PLOS ONE formatting guidelines, we kept the full figure title and description in the figure caption only. “Figure legends”

---

## [Decision Letter · Decision Letter 2]

14 Sep 2025

SENSORY TRIAL OF CAMEL MILK POWDER AMONG PASTORALIST COMMUNITIES OF THE SOMALI REGION, ETHIOPIA

PONE-D-25-31077R2

Dear Dr. Ahmed Mohammed Ibrahim,

We’re pleased to inform you that your manuscript has been judged scientifically suitable for publication and will be formally accepted for publication once it meets all outstanding technical requirements.

Kind regards,

Awatif Abid Al-Judaibi, PhD

Academic Editor

PLOS ONE

Reviewer #2:

Reviewers' comments:

Reviewer's Responses to Questions

**Comments to the Author**

Reviewer #2: All comments have been addressed

2. Is the manuscript technically sound, and do the data support the conclusions?

Reviewer #2: Yes

3. Has the statistical analysis been performed appropriately and rigorously?

Reviewer #2: Yes

4. Have the authors made all data underlying the findings in their manuscript fully available?

Reviewer #2: Yes

5. Is the manuscript presented in an intelligible fashion and written in standard English?

Reviewer #2: Yes

Reviewer #2: (No Response)

**Do you want your identity to be public for this peer review?** For information about this choice, including consent withdrawal, please see our Privacy Policy

Reviewer #2: No

---

## [Editor Report · Acceptance letter]

PONE-D-25-31077R2

PLOS ONE

Dear Dr. Ibrahim,

I'm pleased to inform you that your manuscript has been deemed suitable for publication in PLOS ONE. Congratulations! Your manuscript is now being handed over to our production team.

Kind regards,

on behalf of

Professor Awatif Abid Al-Judaibi

Academic Editor

PLOS ONE